# Tensin2 Is a Novel Diagnostic Marker in GIST, Associated with Gastric Location and Non-Metastatic Tumors

**DOI:** 10.3390/cancers14133212

**Published:** 2022-06-30

**Authors:** Sami Salmikangas, Tom Böhling, Nanna Merikoski, Joanna Jagdeo, Mika Sampo, Tiina Vesterinen, Harri Sihto

**Affiliations:** 1Department of Pathology, University of Helsinki and Helsinki University Hospital, FI-00014 Helsinki, Finland; tom.bohling@helsinki.fi (T.B.); nanna.merikoski@helsinki.fi (N.M.); joanna.jagdeo@helsinki.fi (J.J.); harri.sihto@helsinki.fi (H.S.); 2Department of Pathology, HUSLAB, HUS Diagnostic Center, University of Helsinki and Helsinki University Hospital, FI-00029 Helsinki, Finland; mika.sampo@hus.fi (M.S.); tiina.vesterinen@helsinki.fi (T.V.)

**Keywords:** biomarker, diagnostic marker, gastrointestinal stromal tumor, GIST, immunohistochemistry, sarcoma, Tensin2

## Abstract

**Simple Summary:**

Most of the immunohistochemical markers for gastrointestinal stromal tumor (GIST) are also expressed in other sarcomas relevant for the differential diagnosis of GIST, and thus, new immunohistochemical markers can be of help in GIST diagnostics. Tensin2 (TNS2) was found to be downregulated in most human cancers but overexpressed in GIST; we therefore investigated the role of TNS2 as a diagnostic biomarker for GIST. The results provide conclusive evidence for the value of TNS2 as a sensitive and specific diagnostic biomarker for GIST, with stronger associations for gastric and non-metastatic tumors. These findings can be of use in the differential diagnosis of GIST, especially while trying to differentiate other tumors of the stomach, such as lipo- and leiomyosarcomas, from GISTs.

**Abstract:**

GIST is a rare soft tissue sarcoma, for which KIT and DOG1 are used as highly sensitive diagnostic markers. Other diagnostic markers include CD34, protein kinase C θ, deficiency of succinate dehydrogenase complex subunit B, carbonic anhydrase II, and type I insulin-like growth factor receptor. We investigated the role of TNS2 as a diagnostic biomarker by using immunohistochemistry in 176 GISTs and 521 other sarcomas. All GISTs expressed TNS2, with intermediate or high expression in 71.4% of samples. The majority (89.8%) of other sarcomas were negative for TNS2, and intermediate to strong staining was only seen in 2.9% of samples. Strong TNS2 staining was associated with gastric location (gastric 52.8% vs. non-gastric 7.2%; *p* < 0.001), absence of metastases (non-metastatic tumors 44.3% vs. metastatic tumors 5.9%; *p* = 0.004), female sex (female 45.9% vs. male 33.8%; *p* = 0.029), and tumors of lower risk categories (very low or low 46.9% vs. intermediate 51.7% vs. high 29.0%; *p* = 0.020). TNS2 expression did not correlate with overall survival or metastasis-free survival. No associations between TNS2 expression and KIT/PDGFRA mutation status, tumor size, mitotic count, or age of the patient were detected. The results provide conclusive evidence for the value of TNS2 as a sensitive and specific diagnostic biomarker for GIST.

## 1. Introduction

Gastrointestinal stromal tumor (GIST) is a soft tissue sarcoma with an incidence ranging from 4 to 22 cases per million individuals [1], and yet it is the most common mesenchymal neoplasm of the gastrointestinal (GI) tract. GIST is thought to originate from the interstitial cells of Cajal or their stem cell-like precursors [2], but recently smooth muscle cells were suggested as an alternative cell of origin [3]. Approximately 85–90% of GISTs in adults arise due to activating mutation in either the receptor tyrosine kinase KIT or platelet-derived growth factor receptor α (PDGFRA) genes [4]. The mutations cause a continuous, ligand-independent activation of downstream signaling pathways, leading to increased proliferation and evasion of apoptosis. KIT and PDGFRA are precision medicine targets for receptor tyrosine kinase inhibitors such as imatinib, sunitinib, and regorafenib. The remaining 10–15% of adult cases and up to 90% of pediatric cases that do not contain KIT or PDGFRA mutations usually have mutations in either the RAS pathway [5], the succinate dehydrogenase complex [6], or none of the above.

In addition to being the primary oncogene in GISTs, KIT (CD117) is a highly sensitive diagnostic marker, being expressed in >95% of cases, but it is rarely also expressed in other abdominal or GI tumors relevant for differential diagnosis such as intestinal metastatic melanoma, colorectal adenocarcinoma, angiosarcoma, and extraskeletal Ewing sarcoma [7,8]. Another diagnostic marker for GISTs, DOG1 (Discovered on GIST 1, also known as Anoctamin-1), shows even higher sensitivity, detecting up to 36% of KIT-negative GISTs [9], but it is commonly expressed in gastric carcinomas and mesenchymal sarcomas such as leiomyoma and synovial sarcoma [10]. In a study of 1168 GISTs by Miettinen et al. [7], only 2.6% of GI-tract GISTs were negative for both KIT and DOG1. Other immunohistochemical markers for GIST include CD34 [11], protein kinase C θ (PKC θ) [12], carbonic anhydrase II [13], insulin-like growth factor 1 receptor (IGF-1R) [14], and deficiency of succinate dehydrogenase complex subunit B (SDHB) [15]; however, none of these markers provide very high specificity for GISTs among other sarcomas.

Tensin2 (TNS2, C1-Ten, TENC1) is a member of the tensin family, which is a group of four proteins (Tensin1, Tensin2, Tensin3, and CTEN) important in forming and maintaining the focal adhesions of the cell [16]. All proteins of the tensin family contain the protein tyrosine phosphatase (PTP), Src homology 2 (SH2), and phosphotyrosine-binding (PTB) domains, except for CTEN, which lacks the PTP domain, but the protein kinase C conserved region 1 (C1) domain is found only in TNS2 [17]. Tensins are thought to link the actin filaments of the focal adhesions to the extracellular matrix by binding actin via their N-terminus [16] and integrins via their C-terminal PTB domain [18]. TNS2 has also been shown to regulate important intracellular signaling such as the PI3K-Akt-PKB pathway [19] and the IRS1-Glut4/PDK1 glucose metabolism pathway [20,21]. The role of TNS2 in tumorigenesis is quite unclear, as it has been reported to promote cell migration [22] and proliferation [23] but also to contribute to the tumor-suppressive function via binding to the PTB domain of deleted in liver cancer 1 (DLC1) protein [24]. Recently, Hong et al. [17] demonstrated that TNS2 is downregulated in most human cancers, and low expression is associated with poor survival and silencing of TNS2-promoted tumor growth in HeLa cell-xenografted NSG mice. 

To our knowledge, TNS2 expression has not been used as a diagnostic biomarker in any cancer yet. In this study, we set out to explore the diagnostic value of TNS2 in GIST and its association with clinicopathological parameters such as metastases, mutation status, and tumor location. Here, we present TNS2 as a novel immunohistochemical diagnostic marker for GISTs, displaying both high sensitivity and superior specificity to GIST among soft tissue sarcomas. By comparing the immunohistochemical TNS2 expression in a GIST patient series against the clinicopathological factors, we found TNS2 expression to be associated with gastric and non-metastatic GISTs.

## 2. Materials and Methods

### 2.1. Patients and Clinical Data

A retrospective GIST series included all 161 formalin-fixed, paraffin-embedded (FFPE) tumor tissue samples identified and retrieved from the Helsinki Biobank (Helsinki, Finland) and the corresponding clinical data collected by the Helsinki Biobank from the patient records of Helsinki University Hospital (HUH, Helsinki, Finland). Patients with a primary GIST tumor who were diagnosed in HUH between 1990 and 2020 and who gave a biobank consent were included in the study. Eight tumors were excluded after histological re-evaluation performed by a pathologist with special expertise in the field of sarcomas (M.S.) due to misdiagnosing, scarcity of tumor tissue, or not being a primary tumor sample. In addition, five patients were lacking clinical information and were thus excluded, resulting in 148 tumors in the final series. Clinical information on treatments was not collected. Tumor risk category assessments were done according to the modified National Institutes of Health (M-NIH) consensus risk criteria [25], not accounting for tumor ruptures as that data was unavailable.

The sarcoma screening series consisted of 548 FFPE primary tumor samples (27 GISTs, 19 fibrosarcomas, 89 liposarcomas, 57 leiomyosarcomas, 220 malignant fibrous histiocytomas, 19 malignant peripheral nerve sheath tumors, 45 myxofibrosarcomas, 41 synovial sarcomas, and 31 sarcomas of unknown primary origin) collected between 1987 and 2012 that were retrieved from the Helsinki Biobank. The 27 GISTs included in this series are independent of the GISTs included in the 148-tumor series described above. Histopathological diagnoses were confirmed by M.S., but no clinical data were collected for the series.

### 2.2. Tissue Microarray Construction and Immunohistochemistry

Three core punches per FFPE sample from the representative tumor regions were used to prepare tissue microarray (TMA) blocks in the Helsinki Biobank. Then, 3.5-µm-thick tissue sections were cut on adhesion microscope slides (Labsolute, Geyer GmbH, Renningen, Germany). GIST series TMA was stained with TNS2, KIT, and DOG1, while the sarcoma screening series was stained only for TNS2.

For the TNS2 staining, the tissue sections were incubated at 56 °C for one hour, deparaffinized in xylene, and rehydrated in a decreasing alcohol gradient using the long deparaffinization program in Tissue-Tek DRS 2000 Automated Slide Stainer (Sakura Finetek Japan, Tokyo, Japan). Endogenous peroxidase activity was blocked by incubating the tissue sections in 1% hydrogen peroxide for 30 min. Antigen retrieval was carried out in pH 6.0 EnVision FLEX Target Retrieval Solution (Dako, CA, USA, Cat#: K8005) in a heat-induced epitope retrieval decloaking chamber (Biocare Medical, CA, USA) at 95 °C for 15 min. The tissue sections were then incubated overnight at 4 °C with the primary antibody (polyclonal rabbit anti-TNS2 antibody, Atlas Antibodies, Bromma, Sweden, Cat#: HPA034659) diluted 1:350 in Immunologic Normal Antibody Diluent (WellMed, Duiven, Netherlands, Cat#: BD09-500). After this, the tissue sections were incubated for 30 min at room temperature with the secondary antibody (1-step detection system rabbit HRP, WellMed, Cat#: R500HRP). Finally, the tissue sections were stained for 7 min at room temperature with ImmPACT DAB Substrate kit (Vector Laboratories, CA, USA, Cat#: SK4105) and counterstained for 3 min at room temperature with Mayer’s Hematoxylin (Dako, Cat#: S3309). The tissue sections were washed two times for 5 min between each incubation with 1 x Tris-buffered saline with 0.1% Tween-20 detergent.

KIT and DOG1 were stained at the pathology laboratory of Helsinki University Hospital (HUSLAB) using a Ventana Benchmark Ultra (Roche/Ventana, Tucson, AZ, USA) automated staining system. The sections were deparaffinized using the automated deparaffinization function of the Ventana platform with EZ-Prep solution (950–100, Roche/Ventana). For the KIT staining, the sections were pre-treated for 64 min in CC1 buffer (950–224, Roche/Ventana), after which the sections were incubated for 40 min in the primary antibody (KIT (CD117) polyclonal antibody, Dako/Agilent #A4502, 1:400 dilution) and detected with the UltraView DAB Detection Kit (760–500, Roche/Ventana). For the DOG1 staining, the sections were pre-treated for 64 min in CC1 buffer, incubated for 32 min in the primary antibody (DOG1 clone K9 antibody, Novocastra #NCL-L-DOG-1, 1:100 dilution), and detected with OptiView DAB IHC Detection Kit (760–700, Roche/Ventana). 

The stained sections were digitalized with a Panoramic scanner (3DHISTECH, Budapest, Hungary) using a 20× objective and viewed using CaseViewer software version 2.4.0.119028 (3DHISTECH). TNS2, KIT, and DOG1 expressions were scored based on staining intensity in a blinded fashion regarding the clinical information. The number of positive cells was not considered in the scoring due to mostly homogenous diffuse staining in the tumor samples. The samples were graded by two researchers (S.S. and T.V.) individually, and noncongruent results were graded again together. The samples were graded as showing either no expression (−), weak expression (+), intermediate expression (++), or strong expression (+++) in tumor cells. In the case of heterogeneous staining, the highest result from three replicate samples on TMA defined the expression level of the tumor. None of the tumors were graded as negative for TNS2 expression; thus, the TNS2 expression grading was +/++/+++. Similarly, none of the tumors in the study were graded as weak for DOG1 expression, rendering the DOG1 expression grading −/++/+++. KIT expression grading remained as −/+/++/+++. TNS2 expression was evaluated according to its diffuse cytoplasmic and membranous staining, while KIT expression was evaluated for diffuse cytoplasmic staining and DOG1 expression was evaluated for diffuse membranous and cytoplasmic staining. The interstitial cells of Cajal in a healthy human colon were used as positive staining control for KIT and DOG1, while glomeruli of a healthy human kidney were used as positive staining controls for TNS2.

### 2.3. RNA Extraction and qPCR

RNA was extracted from two 10-µm-thick FFPE primary tumor tissue sections using the QIASymphony RNA Kit (Qiagen GmbH, Hilden, Germany, Cat#: 931636) and QIASymphony SP instrument (Qiagen) according to the manufacturer’s protocol. Of the extracted RNA, 10 µL was then reverse-transcribed into cDNA with SuperScript IV Vilo Master Mix reagent (Thermo Fisher Scientific Baltics UAB, Vilnius, Lithuania, Cat#: 11756500) according to the manufacturer’s instructions.

TNS2 mRNA levels were quantified from 44 GIST tumors using quantitative real-time PCR (qPCR) with hydrolysis probes and a Bio-Rad CFX-384 Touch Real-Time PCR Detection System instrument (Bio-Rad Laboratories, CA, USA). The samples included in the qPCR were randomly picked in a blinded fashion, without the researcher’s knowledge of the IHC staining results or clinicopathological parameters. TNS2 expression was normalized with GAPDH expression, and all samples were run in triplicate. Of cDNA, 1 µL was amplified in a 10 µL reaction using Lightcycler 480 Probes Master kit (Roche Diagnostics GmbH, Mannheim, Germany, Cat#: 04887301001), hydrolysis probes (probe #41 for TNS2 and probe #60 for GAPDH) from the Universal Probe Library Set (Roche Diagnostics, Cat#: 04683633001), and custom-designed primers. The PCR mixture contained 1 x PCR buffer, 100 nM hydrolysis probe, and 200 nM forward and reverse primers. The primers were designed using the ProbeFinder version 2.53 for Human program at the Universal Probe Library Assay Design Center (www.universalprobelibrary.com; Roche Diagnostics, accessed on 17 December 2020). Sequences of the primers used are detailed in Table 1. The qPCR cycling conditions were as follows: 1. initial denaturation at 95 °C for 10 min, 2. denaturation at 95 °C for 30 s, 3. annealing at 60 °C for 30 s, 4. elongation at 72 °C for 45 s, 5. plate read, 6. repeat steps 2–5 for 45 cycles, and 7. final elongation at 72 °C for 7 min.

Results from the qPCR were analyzed using the ΔΔCT method in CFX Maestro 1.1 (version 4.1.2433.1219, Bio-Rad).

### 2.4. DNA Extraction, PCR, and Sequencing

DNA was extracted from FFPE primary tumor tissue blocks using three core punches per sample, using the QIASymphony DSP DNA Mini Kit (Qiagen, Cat#: 937236) and QIASymphony SP instrument (Qiagen) and eluted into 50 µL of sterile water. PCR for KIT exons 9, 11, 13, and 17 and PDGFRA exons 12 and 18 was performed with FastStart Taq DNA Polymerase dNTPack kit (Roche Diagnostics, Cat#: 04738381001). About 20 to 200 ng of DNA per sample was amplified in a 20 µL reaction in a 96-well format using a PTC-100 thermal cycler (MJ Research, Watertown, MA, USA). The PCR mixture contained 1 × PCR buffer, 20 mM MgCl2, 300 nM forward and reverse primers, 0.2 mM dNTP solution, and 1 U of DNA polymerase. Sequences of the primers used are given in Table 1. The PCR cycling conditions were as follows: 1. initial denaturation at 95 °C for 4 min, 2. denaturation at 95 °C for 30 s, 3. annealing at 60 °C for 30 s, 4. elongation at 72 °C for 30 s, 5. repeat steps 2–4 for 40 cycles, and 6. final elongation at 72 °C for 7 min.

PCR reactions were purified with Applied Biosystems ExoSap-IT PCR Product Cleanup Reagent (Thermo Fisher Scientific, Cat#: 78201.1.ML) according to the manufacturer’s instructions. Amplified DNA samples were screened for mutations in KIT exons 9, 11, 13, and 17 and PDGFRA exons 12 and 18 by sanger-sequencing at the Institute for Molecular Medicine Finland (FIMM, Helsinki, Finland), using an ABI3730xl DNA Analyzer (Thermo Fisher Scientific) according to the manufacturer’s instructions.

### 2.5. Statistical Analyses

Statistical analyses were performed using SPSS statistics version 27.0 (IBM Corporation, New York, NY, USA). Continuous variables (age, tumor size, and mitotic count) were analyzed with a two-sided Kruskal–Wallis (KW) test or Mann–Whitney U (MW) test, and non-continuous variables (sex, tumor location, metastasis status, tumor risk category, and mutation status) were analyzed with a two-sided Fisher–Freeman–Halton Exact (FH) test. Overall and metastasis-free survival rates were estimated with the Kaplan–Meier method and compared with the Mantel–Cox log-rank (MC) test. Overall survival was calculated from the date of GIST diagnosis to the date of death, censoring patients alive on their last follow-up date. Metastasis-free survival was calculated from the date of GIST diagnosis to the date of diagnosis of GIST-related metastasis, excluding patients that had a metastasis at the date of GIST diagnosis and censoring patients who did not have a metastasis diagnosis when the data collection for the study was closed (3 November 2021) or who had died without diagnoses of the metastases. *p*-values of less than 0.05 were considered significant.

## 3. Results

### 3.1. Clinicopathological Data of Patients

Clinicopathological data of the patients in the GIST series are provided in Table 2. Our cohort included 148 GIST patients, with a median age of 65 years at the time of GIST diagnosis. Primary tumor location was gastric in 106 patients (71.6%) and non-gastric in 42 patients (28.4%). Median tumor size was 5.5 cm and median mitotic count was 4.0 per 50 HPF. Nine tumors (6.1%) had metastasized at the time of diagnosis, and another eight (5.4%) had metastasized during the follow-up, resulting in a total of 17 metastases (11.5%). KIT exon 9, 11, or 13 mutations were found in 103 tumors (71.5%), PDGFRA exon 12 or 18 mutations were found in 19 tumors (13.2%), and 22 tumors (15.3%) did not have mutations in either the KIT or the PDGFRA gene (KIT/PDGFRA WT). No KIT exon 17 mutations were found.

### 3.2. TNS2 Is Highly Overexpressed in GIST Compared with Other Cancers

First, we visually investigated expression level of TNS2 in an IST transcriptome database (https://ist.medisapiens.com/, accessed on 15 January 2021) of MediSapiens Ltd (Helsinki, Finland). [26]. The database currently consists of gene expression array transcriptomes of nearly 20,000 tissue samples from about 400 tissue types, including 77 GIST samples. TNS2 expression in GISTs was relatively close to the level of expression in healthy tissues but overexpressed in GISTs compared with most other cancers (Figure 1; Appendix A, Figure A1). The median of relative expression of TNS2 is more than doubled in GISTs compared with most other cancers included in the IST Online database.

### 3.3. TNS2 Is a Potent Diagnostic Marker for GIST

The GIST series (*n* = 148) was stained for all three markers (TNS2, KIT, and DOG1), while the sarcoma screening series (*n* = 548) was only stained for TNS2. Examples of the immunohistochemical stainings are shown in Figure 2.

To validate the immunohistochemistry results, TNS2 mRNA expression was investigated by qPCR in 44 GIST samples, of which 12 displayed weak, 13 intermediate, and 19 strong TNS2 staining. The correlation between TNS2 mRNA expression and TNS2 protein expression was strong (Figure 3; *p* < 0.001, KW test). Mean relative gene expression was 9.7 for weak samples (95% CI: 4.7–14.7), 35.3 for intermediate samples (95% CI: 18.7–51.8), and 57.2 for strong samples (95% CI: 29.0–85.4).

TNS2 was highly expressed in GISTs (71.4% intermediate or strong expression) compared with other sarcomas (2.9% intermediate or strong expression) (Table 3). A total of 70 GISTs (40%) showed strong TNS2 expression, whereas none of the other sarcoma types showed strong expression. All 175 GISTs examined displayed TNS2 expression, while 468 (89.8%) of the 521 other sarcomas were completely negative for TNS2. 

A positive predictive value (PPV) and negative predictive value (NPV) to distinguish GISTs from other sarcomas based on TNS2 immunohistochemical expressions were calculated from the sarcoma screening series. The prevalence of GISTs in the sarcoma screening series was 4.9% (27 of 548). When the negative staining was compared to positive staining (weak, intermediate, or strong), the PPV was 33.8% and the NPV was 100%. When the negative and weak stainings were compared to the intermediate and strong stainings, the PPV was 59.5% and NPV was 97.8%.

### 3.4. Sensitivity of TNS2 as a Diagnostic Marker Compared with KIT and DOG1

The expression of TNS2, KIT, and DOG1 was categorized as negative (−) or positive (+/++/+++) in the series of 148 GISTs. All 148 tumors were positive for TNS2, and 146 and 147 were positive for DOG1 or KIT, respectively. Only one tumor was negative for both KIT and DOG1.

### 3.5. Immunohistochemical TNS2 Expression Is Stronger in Gastric and Non-Metastatic GISTs

Immunohistochemical expression of TNS2 was compared with sex, age at diagnosis, primary tumor location, tumor size, mitotic count, metastasis status, tumor risk category, and mutation status (Table 4). TNS2 expression was found to be stronger in gastric tumors than in non-gastric tumors (*p* < 0.001, FH test) and in non-metastatic tumors than in metastatic tumors (*p* = 0.004, FH test). TNS2 expression was also found to be lower in tumors of the high-risk category than in tumors of other risk categories (*p* = 0.020, FH test). In addition, the tumors of female patients had stronger TNS2 expression than the tumors of male patients (*p* = 0.029, FH test). Age, tumor size, mitotic count, and mutation status did not differ among the different TNS2 expression groups (all *p*-values > 0.05, KW or FH test). The associations remained the same when examining weak TNS2 expression vs. intermediate and strong TNS2 expression (data not shown).

### 3.6. TNS2 Expression Is Not Associated with Overall or Metastasis-Free Survival

Median follow-up times after GIST diagnosis for patients of different TNS2 expression levels who were alive were 7.6 years for weak (*n* = 30, range 1.5–15.3 years), 7.5 years for intermediate (*n* = 29, range 0.1–16.7 years), and 7.1 years for strong (*n* = 42, range 0–15.7 years). A total of 47 patients (31.8%) died during follow-up. No association was found between TNS2 expression level and overall survival (*p* = 0.610, MC test) (Figure 4, panel A). 

Median follow-up times after GIST diagnosis for patients of different TNS2 expression levels whose tumors did not develop metastases were 6.6 years for weak (*n* = 36, range 0.1–15.3 years), 6.3 years for intermediate (*n* = 37, range 0.1–17.3 years), and 8.6 years for strong (*n* = 58, range 0.1–19.6 years). A total of 9 patients (6.1%) had metastases at the time of diagnosis, and 8 patients (5.4%) developed a metastasis after the diagnosis. No association was found between TNS2 expression level and metastasis-free survival (*p* = 0.190, MC test) (Figure 4, panel B). No association with overall or metastasis-free survival was found when examining weak TNS2 expression vs. intermediate and strong TNS2 expression (data not shown).

## 4. Discussion

By visually investigating the expression level of TNS2 in a transcriptome database, we found that the expression of TNS2 in GIST was close to that of healthy tissues, but highly overexpressed compared with other cancers. Hong et al. [17] have also reported TNS2 to be downregulated in most cancers, which then led us to investigate the possibility of TNS2 being a diagnostic marker for GIST. Here, we provide conclusive evidence that TNS2 is indeed a promising diagnostic marker for GIST, being specific to GIST among other sarcomas.

The differential diagnosis of gastric GIST can be difficult at times due to many non-GIST sarcomas, such as liposarcomas, leiomyosarcomas, and unclassified sarcomas, being able to develop in the stomach [27]. KIT and DOG1 are used as highly sensitive diagnostic markers for GIST, but they are expressed also in other mesenchymal tumors relevant for GIST differential diagnosis, such as synovial sarcomas, leiomyomas, leiomyosarcomas, angiosarcomas, Ewing sarcomas, malignant peripheral nerve sheath tumors, and schwannomas [28]. However, gastric adenocarcinomas do not provide a differential diagnostic problem, as they are most usually negative for KIT, DOG1, and CD34, while GISTs are usually positive for at least one of these markers [29]. In the present study, we found that 89% of gastric GISTs have intermediate to strong TNS2 expression, while 90% of non-GIST sarcomas examined were completely negative for TNS2 expression. These results could be helpful in diagnostics to better differentiate GISTs from other gastric sarcomas. Other mesenchymal gastric tumors include lipomas, glomus tumors, hemangiomas, inflammatory fibroid polyps, and inflammatory myofibroblastic tumors [27]. In addition to strong TNS2 expression, gastric GISTs are associated with high IGF1-R expression [14], mutations in PDGFRA instead of KIT [30], and a better overall prognosis [30].

We found TNS2 expression to be lower in high-risk category tumors than in tumors of other risk categories. However, almost all intermediate-risk category tumors were gastric in the series due to the modified NIH risk stratification criteria, and thus, have inherently stronger TNS2 expression. These results indicate that strong TNS2 expression could act as a marker for lower risk of recurrence in GIST, but further studies are required to confirm this.

TNS2 expression was found to be stronger in female patients’ tumors than in male patients’ tumors. Recently, Rong et al. [31] showed male gastric GIST patients to have a higher mortality rate than female patients, and the survival of female gastric GIST patients was better than that of male patients, with sex being an independent risk factor for overall survival. Despite TNS2 expression being stronger in gastric tumors and in female patients in our cohort, we found no association between TNS2 expression and overall survival. An earlier study also displayed sex as a prognostic factor for GIST [32], while in another study sex had no significant prognostic effect [33]. We found no association between sex and overall survival in gastric GIST in our cohort (data not shown). The disparity between results might be explained by the relatively low sample sizes or differing characteristics of patient populations, such as region, culture, race, and lifestyle, since different sexes have very different characteristics around the world.

The functional role of TNS2 remains unclear, as in some cancers, such as hepatocellular carcinoma, it has been shown to promote aggressive tumor behavior [22,23], but it has also been established to have tumor-suppressive properties and functions [17,24]. TNS2 has been shown to dephosphorylate the insulin receptor substrate 1 (IRS-1) and decrease its activation [20], which in turn can lead to increased interaction between PI3K and PDGFRA, and thus, increased tumor growth [34]. As PDGFRA is the most usually mutated driver gene in KIT-wildtype GISTs, this possibility of the suggested TNS2¬–IRS-1–PI3K–PDGFR pathway existing in GIST is intriguing. Another possible tumor-driving pathway with TNS2 includes AXL, IRS-1, and GLUT4. AXL has been reported to be upregulated in KIT-dead GISTs [35], while also binding and phosphorylating TNS2 [21]. This releases TNS2 from its binding partner IRS-1, which in turn upregulates Glut4 and PDK1, enzymes that may play a critical role in the glucose metabolism of cancer cells. In fact, IGF-1R, the upstream receptor of IRS-1, has been suggested to be a potential therapeutic target in GIST [36], in addition to being a diagnostic marker for non-intestinal GIST [14]. Possible TNS2-mediated signaling in GIST involving the IGF-1R–IRS-1 and PI3K-AKT pathways seems to be a promising prospect for future studies.

As a focal adhesion protein bridging the extracellular matrix and the intracellular actin cytoskeleton, TNS2 is distinctively linked to cell motility by either inhibiting migration [19] or promoting it [22]. The direction of the effect that TNS2 has on cell motility most likely depends on the cellular context, level of expression, or differing isoform expression. According to our results, TNS2 expression was associated with non-metastatic GISTs but not with metastasis-free survival. This might be due to the small number of tumors (*n* = 8) that disseminated during the follow-up. In renal cell cancer, TNS2 was shown to not correlate with metastases to nearby lymph nodes or to infiltrate to blood and lymphatic vessels [37]. Many other cell adhesion molecules, including members of the Tensin family but not TNS2, have been linked to metastatic activity in cancers [38]. However, TNS2 expression has been shown to correlate with the malignancy grade of gastric cancer, in addition to being more abundant in moderately differentiated tumors compared to poorly differentiated- or non-differentiated tumors, and correlating with peritumoral inflammation and *H.pylori* infection [39]. Interestingly, TNS2 expression in gastric cancer did not correlate with sex, tumor localization, or metastases, unlike TNS2 expression in GIST in this study.

While screening the tumors for mutations in the KIT and PDGFRA genes, we found the ratios of mutations to be mostly in line with the previous literature [40]. TNS2 expression did not significantly differ among the different mutated genes. Tumors that had a PDGFRA mutation were gastric, which is typical for PDGFRA-mutated GISTs, but TNS2 expression was not stronger in PDGFRA-mutated tumors than in KIT-mutated tumors or KIT/PDGFRA wildtype tumors. 

Additionally, we found one very rare KIT exon 13 point mutation, changing the histidine at codon 630 to tyrosine (KIT nucleotide 1888 C > T, codon H630Y), while other investigated exons were wild types. This mutation is within the protein kinase domain of the KIT protein and has been identified before in a metastatic non-small-cell lung cancer [41] and a vocal cord mucosal melanoma [42], but to our knowledge, this is the first time encountering this mutation in a GIST. The clinical relevance of this mutation in GIST is unknown.

The tumor histologies were re-evaluated by a professional pathologist with a special interest in the field of sarcomas, and the IHC stainings were assessed by two researchers individually, increasing the reliability of the study. On the other hand, the small number of metastases in the cohort limited our ability to further examine the relationship between TNS2 and metastases. It is important to remember that the primary antibody used for TNS2 staining is not for diagnostic use, and as such, the results might not be as reliable as with an established diagnostic antibody. However, since our qPCR results correlated with our IHC results, it is safe to say that the specificity of the antibody was reliable. It is also good to remember the importance of staining optimization in clinical work. TMAs were used in the study due to the large number of samples to be stained and the scarcity of tumor tissue in the samples. The stainings for TNS2 were heterogeneous in some tumors, but as we had three replicate TMA spots in each tumor, the risk of misevaluation of expression levels in tumors was minimized. Due to possible interlaboratory differences in tissue fixation, further study of TNS2 expression in an independent validation cohort would be beneficial.

The development of imatinib, along with other receptor tyrosine kinase inhibitors, has drastically improved the overall survival and the metastasis-free survival of patients with advanced, metastatic, and/or operable GISTs. Glivec (imatinib mesylate), the standard first-line treatment of metastatic GIST, was granted marketing authorization by the European Medicines Agency on 7 November 2001 (agency product number EMEA/H/C/000406), after which other tyrosine kinase inhibitors have followed. The use of these drugs could have affected our patient cohort’s overall and metastasis-free survival results by raising the survival in patients diagnosed after this date since our cohort included patients with a GIST diagnosis between 1990 and 2020. However, only 4 of our 148 patients were diagnosed before 7 November 2001, and thus, the possibility that the rapidly spread use of targeted therapeutic agents could have affected our results is quite small.

The role of TNS2 in GIST tumorigenesis or tumor suppression has not been researched yet, and due to being overexpressed in GISTs compared with other cancers, warrants functional studies on its possible tumor-driving or tumor-suppressing roles and modes of action. Evidence of TNS2′s role in the metastatic activity of tumors is scarce and should be further investigated. Additionally, future studies on the expression of TNS2 in other gastrointestinal malignancies, such as lipomas and schwannomas, could be beneficial to more accurately define the role of TNS2 as a diagnostic marker.

## 5. Conclusions

TNS2 is a promising sensitive and specific novel biomarker for GIST and could be included in the palette of GIST diagnostic markers. TNS2 mRNA expression is higher in GIST than in any other cancer, and is also overexpressed on the protein level compared with other sarcomas. The differential diagnosis of GIST can occasionally be difficult due to tumors such as lipo- and leiomyosarcomas being able to form in the stomach. This is where TNS2 can provide value in diagnostics since strong TNS2 expression was associated with gastric location. TNS2 expression was also associated with lack of metastases, female sex, and lower risk category, and this could be helpful in identifying and classifying tumors of these characteristics.

## Figures and Tables

**Figure 1 cancers-14-03212-f001:**
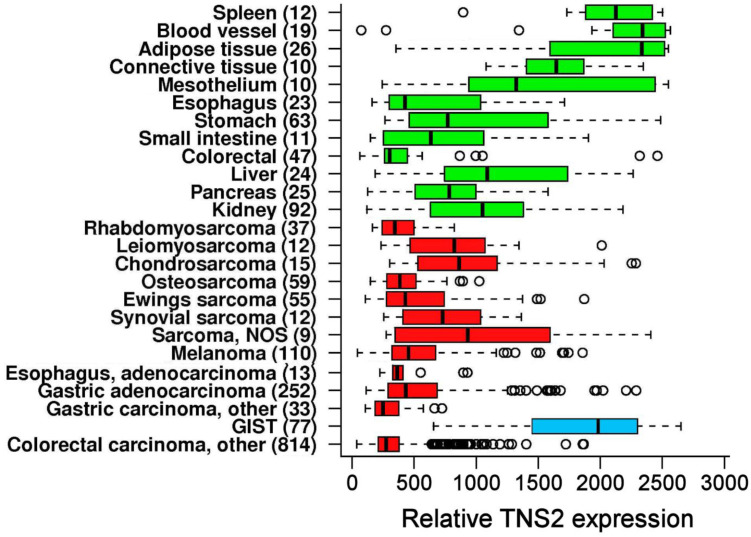
Relative expression of TNS2 among different selected healthy (green) and cancerous (red) tissues. Expression in GIST is highlighted in blue. Numbers of cases are given in parentheses and individual outlier samples are shown as hollow circles. Data are modified from the IST database, Medisapiens Ltd. (https://ist.medisapiens.com/, accessed on 15 January 2021).

**Figure 2 cancers-14-03212-f002:**
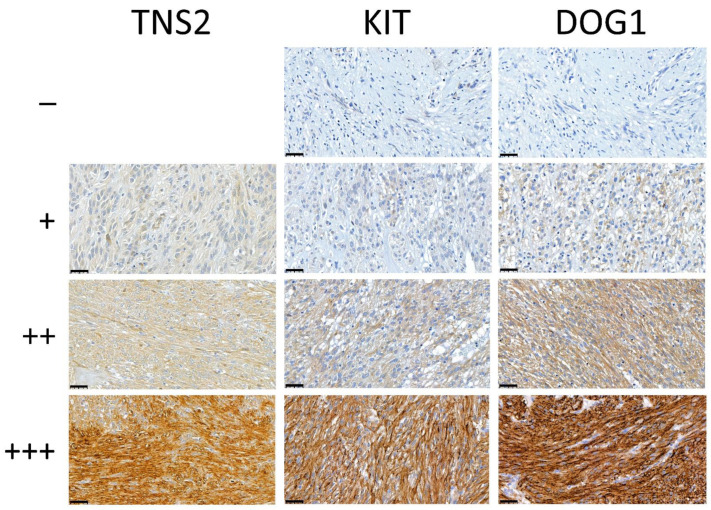
Representative images of TNS2, KIT, and DOG1 immunohistochemical stainings in GIST TMA samples, graded as negative (−), weak (+), intermediate (++), and strong (+++) (from **top** to **bottom**). None of the TMA spots stained negative for TNS2. The pictures are 400× magnifications of the TMA samples. Scale bars in the lower left corners are 40 µm.

**Figure 3 cancers-14-03212-f003:**
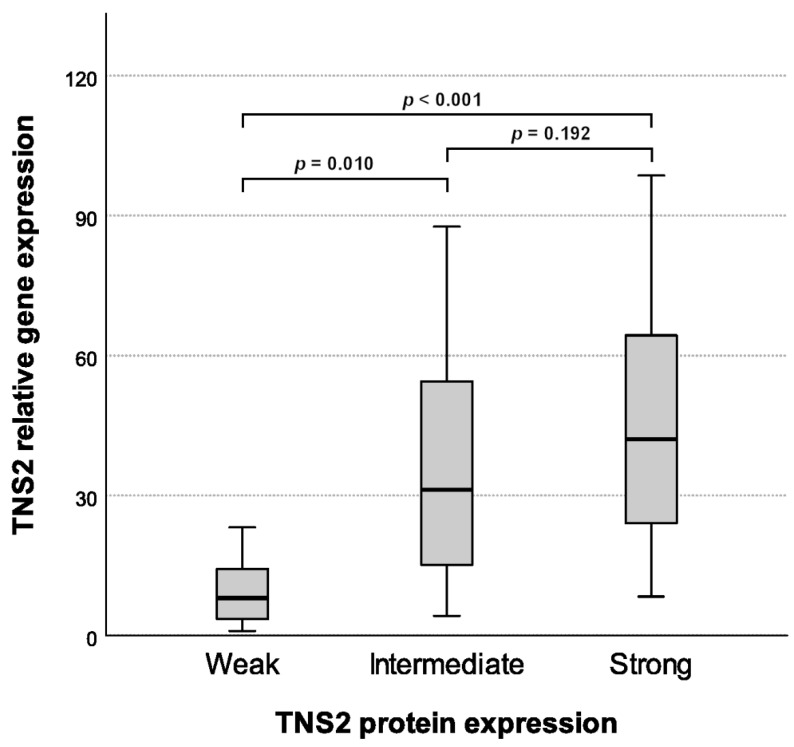
Relative TNS2 gene expression compared with immunohistochemical TNS2 protein expression. Medians are presented as black lines within interquartile ranges. TNS2 relative gene expression positively correlated with the TNS2 protein expression (*p* < 0.001, KW test). *p*-values of pairwise comparisons (weak-intermediate *p* = 0.010, weak-strong *p* < 0.001, intermediate-strong *p* = 0.192) are presented on the top.

**Figure 4 cancers-14-03212-f004:**
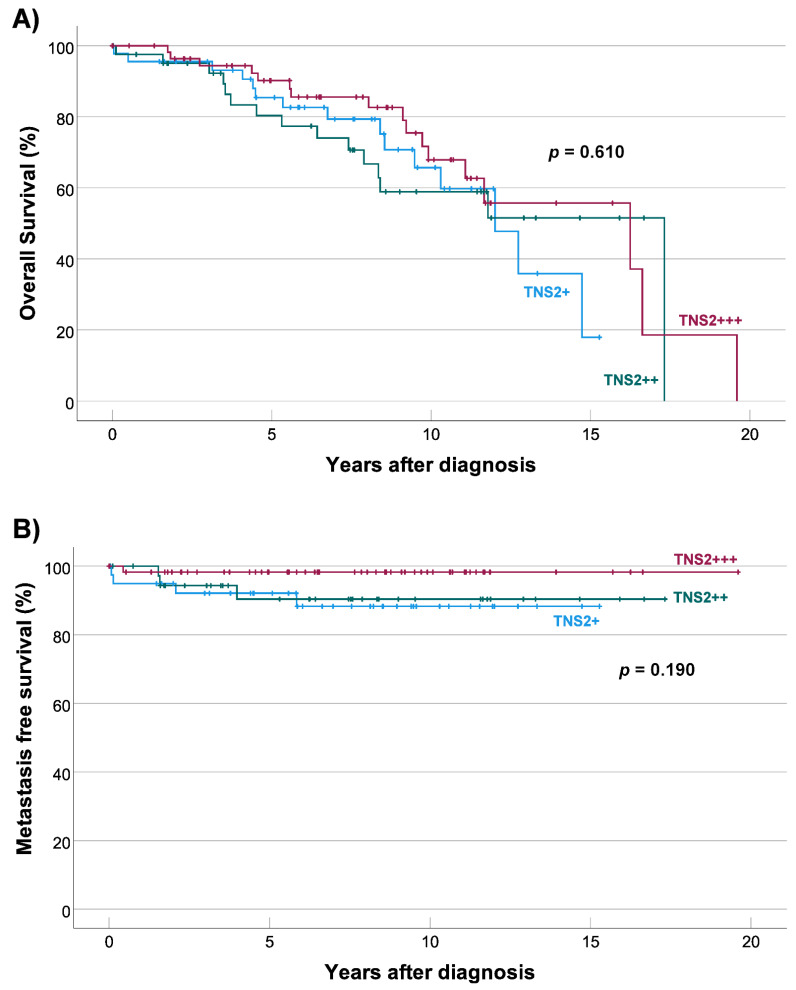
Kaplan–Meier analyses of overall survival (**A**) and metastasis-free survival (**B**) for patients with weak (TNS2+), intermediate (TNS2++), or strong (TNS2+++) expression. (**A**) No association was found between overall survival and TNS2 expression level (*p* = 0.610, MC test). (**B**) No association was found between metastasis-free survival and TNS2 expression level (*p* = 0.190, MC test).

**Table 1 cancers-14-03212-t001:** Primers and probes used in PCR and quantitative PCR.

Target	Forward Primer	Reverse Primer	qPCR Probe
TNS2	5′ -AC AGA AAA TGT GAA GCA AAG GT- 3′	5′ -GTG CTC TAT GCG CCT GAC T- 3′	Universal Probe Library #41
GAPDH	5′ -AGC CAC ATC GCT CAG ACA C- 3′	5′ -GCC CAA TAC GAC CAA ATC C- 3′	Universal Probe Library #60
KIT exon 9	5’-AGC CAG GGC TTT TGT TTT CT-3’	5’-CAG AGC CTA AAC ATC CCC TTA-3’	
KIT exon 11	5’-TTT CCC TTT CTC CCC ACA G-3’	5’-CCC AAA AAG GTG ACA TGG A-3	
KIT exon 13	5’-TAC TGC ATG CGC TTG ACA TC-3	5’-CAT GTT TTG ATA ACC TGA CAG ACA-3	
KIT exon 17	5’-TGG TTT TCT TTT CTC CTC CAA-3	5’-TCA CAG GAA ACA ATT TTT ATC GAA-3	
PDGFRA exon 12	5’-TCC AGT CAC TGT GCT GCT TC-3’	5’-GGA GGT TAC CCC ATG GAA CT-3	
PDGFRA exon 18	5’-CTT GCA GGG GTG ATG CTA TT-3’	5’-TGA AGG AGG ATG AGC CTG AC-3	

**Table 2 cancers-14-03212-t002:** Clinicopathological data of the 148 GIST patient series.

Characteristic	*n* = 148 (%)
Sex	
Male	74 (50.0)
Female	74 (50.0)
Age	Range 10–91
≤50	19 (12.8)
51–60	37 (25.0)
61–70	46 (31.1)
71–80	39 (26.4)
≥81	7 (4.7)
Primary tumor location	
Gastric	106 (71.6)
Small intestinal	33 (22.3)
Large intestinal	2 (1.4)
Extragastrointestinal	7 (4.7)
Tumor diameter in cm	Range 0.9–29.0
≤2.0	6 (4.1)
2.1–5.0	61 (42.1)
5.1–10.0	54 (37.2)
>10.0	24 (16.6)
Not available	3
Cell mitoses per 50 HPF	Range 0–100
≤5	90 (65.7)
6–10	21 (15.3)
>10	26 (19.0)
Not available	11
Risk category	
Very low	4 (2.9)
Low	45 (32.1)
Intermediate	29 (20.7)
High	62 (44.3)
Not available	8
Metastasis status	
Metastatic at diagnosis	9 (6.1)
Metastatic after diagnosis	8 (5.4)
Total metastatic	17 (11.5)
Not metastatic	131 (88.5)
Not available	8
Mutated gene and exon	
KIT exon 9	11 (7.6)
KIT exon 11	86 (59.7)
KIT exon 13	6 (4.2)
PDGFRA exon 12	2 (1.4)
PDGFRA exon 18	17 (11.8)
KIT/PDGFRA WT	22 (15.3)
Not available	4

**Table 3 cancers-14-03212-t003:** TNS2 immunohistochemical expression in 175 GISTs and 521 other soft tissue sarcomas, graded as negative, weak, intermediate, or strong expression.

Sarcoma	*n*	Negative (%)	Weak (%)	Intermediate (%)	Strong (%)
Fibrosarcoma	19	19 (100)	0 (0)	0 (0)	0 (0)
Liposarcoma	89	77 (86.5)	7 (7.9)	5 (5.6)	0 (0)
Leiomyosarcoma	57	50 (87.7)	5 (8.8)	2 (3.5)	0 (0)
Malignant fibrous histiocytoma	220	200 (90.9)	15 (6.8)	5 (2.3)	0 (0)
Malignant peripheral nerve sheath tumor	19	17 (89.5)	2 (10.5)	0 (0)	0 (0)
Myxofibrosarcoma	45	37 (82.2)	6 (13.3)	2 (4.5)	0 (0)
Sarcoma of unknown primary (NUD)	31	28 (90.3)	3 (9.7)	0 (0)	0 (0)
Synovial sarcoma	41	40 (97.6)	0 (0)	1 (2.4)	0 (0)
**Total other sarcomas**	**521**	**468 (89.8)**	**38 (7.3)**	**15 (2.9)**	**0 (0)**
Sarcoma series; GIST	27	0 (0)	5 (18.6)	11 (40.7)	11 (40.7)
GIST series	148	0 (0)	45 (30.4)	44 (29.7)	59 (39.9)
**Total GISTs**	**175**	**0 (0)**	**50 (28.6)**	**55 (31.4)**	**70 (40.0)**

**Table 4 cancers-14-03212-t004:** TNS2 immunostaining and clinicopathological features of the 148 GIST patient sample series.

Characteristic	TNS2 Expression			*p*-Value
	**Weak (*n* = 45)**	**Intermediate** **(*n* = 44)**	**Strong (*n* = 59)**	
Sex, *n* (%)				0.029
Male (*n* = 74)	30 (40.5)	19 (25.7)	25 (33.8)	
Female (*n* = 74)	15 (20.3)	25 (33.8)	34 (45.9)	
Median age at diagnosis (range), years	59.0 (10–91)	65.5 (44–87)	66.0 (35–89)	0.131
Primary tumor location, *n* (%)				<0.001
Gastric (*n* = 106)	14 (13.2)	36 (34.0)	56 (52.8)	
Non-gastric (*n* = 42)	31 (73.8)	8 (19.0)	3 (7.2)	
Median diameter of primary tumor (range), cm	6.0 (2.0–25.0)	5.8 (2.0–29.0)	5.0 (0.9–19.0)	0.197
Not available	1	0	2	
Median amount of cell mitoses per 50 HPF (range)	5.0 (1–80)	4.0 (0–100)	4.0 (0–45)	0.260
Not available	4	2	5	
Metastatic GIST tumor, *n* (%)				0.004
Not metastatic (*n* = 131)	36 (27.5)	37 (28.2)	58 (44.3)	
Metastatic (*n* = 17)	9 (52.9)	7 (41.2)	1 (5.9)	
Risk category, *n* (%)				0.020
Very low/Low (*n* = 59)	13 (26.5)	13 (26.5)	23 (46.9)	
Intermediate (*n* = 29)	3 (10.3)	11 (37.9)	15 (51.7)	
High (*n* = 62)	26 (41.9)	18 (29.0)	18 (29.0)	
Not available	3	2	3	
Mutated gene, *n* (%)				0.442
KIT (*n* = 103)	33 (32.0)	26 (25.2)	44 (42.7)	
PDGFRA (*n* = 19)	4 (21.1)	9 (47.4)	6 (31.6)	
KIT/PDGFRA WT (*n* = 22)	7 (31.8)	7 (31.8)	8 (36.4)	
Not available	1	2	1	

## Data Availability

Restrictions apply to the availability of the transcriptome data of IST Online database. Data was obtained from MediSapiens Ltd. and are available [https://ist.medisapiens.com/] with the permission of MediSapiens Ltd. GIST tissue samples and clinical data were obtained from Helsinki Biobank and are available [https://www.helsinginbiopankki.fi/] with the permission of Helsinki Biobank.

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
