# Peer review of "Tensin2 Is a Novel Diagnostic Marker in GIST, Associated with Gastric Location and Non-Metastatic Tumors"

_cancers, 2022, doi:10.3390/cancers14133212_

Round 1

Reviewer 1 Report

The authors aim to assess the diagnostic role of TNS2 expression in a cohort of GISTs. This is a novel and interesting topic, and the manuscript is overall well written. Only a few minor issues need to be addressed:

1. Section 2.2: it seems that IHC assessment is based on staining intensity only, so the number of positive cells has not been included in such "expression grading", probably due to the presence of diffuse staining in all cases, regardless of its intensity (as far as one can see): please clarify and justify this choice

2. In order to stress the diagnostic potential of TNS2 in distinguishing between GISTs and other sarcomas, I suggest to calculate positive and negative predictive values

Author Response

We thank the reviewer for their valuable and helpful comments. Point-by-point comments and responses are included below. One additional change was made; Figure 3 has been remade to better represent the distribution of relative gene expressions in our samples.

The authors aim to assess the diagnostic role of TNS2 expression in a cohort of GISTs. This is a novel and interesting topic, and the manuscript is overall well written. Only a few minor issues need to be addressed:

  1. Section 2.2: it seems that IHC assessment is based on staining intensity only, so the number of positive cells has not been included in such "expression grading", probably due to the presence of diffuse staining in all cases, regardless of its intensity (as far as one can see): please clarify and justify this choice

RE: Indeed, diffuse staining was an issue in categorizing tumor cells as positive or negative. We have now clarified and justified our grading in the section 2.2. as follows (lines 152-154): “The number of positive cells was not considered in the scoring due to mostly homogenous diffuse staining in the tumor samples.”

  1. In order to stress the diagnostic potential of TNS2 in distinguishing between GISTs and other sarcomas, I suggest to calculate positive and negative predictive values

RE: Positive and negative predictive values were calculated. We have added these results under the section 3.3 (lines 284-290).

Reviewer 2 Report

In the present manuscript, the authors attempt to show that Tensin2 (TNS2) can be used as a sensitive and specific diagnostic biomarker for gastrointestinal stromal tumor (GIST). They show that all GISTs expressed TNS2, with intermediate or high expression in 71.4% of samples and the majority (89.8%) of other sarcomas were negative for TNS2, and intermediate to strong staining was only seen in 2.9% of samples. Besides, Strong TNS2 staining was associated with gastric location, absence of metastases, sex, and tumors of lower risk categories. Thus, in principle, this manuscript addresses a relevant issue. Unfortunately, the results presented in the report reviewed here are far from convincing.

Some specific points:

1. Please carefully check the number of samples used in the screening series.

2.Table 2 only gives a simple description of the data without more in-depth data analysis.

3.In most of the figures, the authors did not label and explain statistical significance (P value) in the charts.

4.Line 236-237, it is difficult to draw such a conclusion from the results in Figure 1. At the same time, such a conclusion is independent of the authors' goal of proving that TNS2 can be used as a diagnostic biomarker for GIST.

5.The Figure 2 is difficult to present as an experimental result and can be used as a standard for staining grade evaluation in the method.

6.Line 254-257, Please describe the random sampling method in detail in the method. Here, I recommend that all 148 specimens be tested by qPCR. I think the mRNA results can be put at the end of the results

7.In the title of Table 4, the number of GIST is 149.

8.The validation cohort should be introduced into the research results.

Author Response

We thank the reviewer for their valuable and helpful comments. Point-by-point comments and responses are included below. One additional change was made; Figure 3 has been remade to better represent the distribution of relative gene expressions in our samples.

In the present manuscript, the authors attempt to show that Tensin2 (TNS2) can be used as a sensitive and specific diagnostic biomarker for gastrointestinal stromal tumor (GIST). They show that all GISTs expressed TNS2, with intermediate or high expression in 71.4% of samples and the majority (89.8%) of other sarcomas were negative for TNS2, and intermediate to strong staining was only seen in 2.9% of samples. Besides, Strong TNS2 staining was associated with gastric location, absence of metastases, sex, and tumors of lower risk categories. Thus, in principle, this manuscript addresses a relevant issue. Unfortunately, the results presented in the report reviewed here are far from convincing.

Some specific points:

  1. Please carefully check the number of samples used in the screening series.

RE: The error has now been fixed (line 105: “554 samples” > “548 samples”).

2.Table 2 only gives a simple description of the data without more in-depth data analysis.

RE: This is as intended; Table 2 is supposed to describe the clinicopathological characteristics of our GIST cohort. Table 4 has a more in-depth analysis of the relationship of the clinicopathological characteristics and TNS2 expression.

3.In most of the figures, the authors did not label and explain statistical significance (P value) in the charts.

RE: P-values have been added to figures 3 and 4.

4.Line 236-237, it is difficult to draw such a conclusion from the results in Figure 1. At the same time, such a conclusion is independent of the authors' goal of proving that TNS2 can be used as a diagnostic biomarker for GIST.

RE: More healthy tissues were added to Figure 1 to make the conclusion more convincing. The Appendix A, Figure A1 represents the relative TNS2 expression results in all investigated tissue types, but due to the large number of different tissue types, it is not a reader friendly figure and we decided to present only a subset of the results in the Figure 1.

5.The Figure 2 is difficult to present as an experimental result and can be used as a standard for staining grade evaluation in the method.

RE: The Figure 2 is a direct example of results from our IHC experiment, and thus we would like to keep it in the results section.

6.Line 254-257, Please describe the random sampling method in detail in the method. Here, I recommend that all 148 specimens be tested by qPCR. I think the mRNA results can be put at the end of the results

RE: The samples in the cohort were given a running number from 1-148, and simply the first 44 samples from the list were included in the qPCR experiment without the researcher’s knowledge of staining intensities or clinicopathological characteristics. This has now been described in the methods (lines 177-179). The difference in the relative gene expression between weakly, intermediately, and strongly stained samples was statistically very significant (p < 0.001), even with the 44 samples. As such, we don’t believe that replicating the experiment with all 148 specimens will change the results in any way.

7.In the title of Table 4, the number of GIST is 149.

RE: This error has now been fixed.

8.The validation cohort should be introduced into the research results.

RE: We agree that the independent validation cohort would improve the manuscript, and added the following line to the Discussion (lines 430-431): “Due to possible interlaboratory differences in tissue fixation, further study of TNS2 expression in an independent validation cohort would be beneficial.” However, the results presented in this manuscript are already from three independent cohorts (the transcriptome data from 77 GISTs presented in Figure 1 and Figure A1, the sarcoma screening series including 27 GISTs, and our main GIST series of 148 samples). Thus, we believe that the use of these three individual cohorts provide representative and conclusive evidence for our claims.

Reviewer 3 Report

It is a well-prepared manuscript containing new scientific findings. Their translation into everyday diagnostics is also promising. However, some clarifications are still needed on certain points for publication.

Detailed comments:

Line 105
sarcoma screening series:
It should be described how this differs from the previous series, as it also includes GISTs. Are these also included in the other series or are they separate cases? What criteria were used to select this series?

Line 117
What was the reason for not adapting TNS2 IHC as an automated procedure to run on the Ventana immunohistochemistry platform?

Line 128
“30 minutes at room temperature in the secondary antibody”
in -> with (I suggest to use “with” instead of “in”)

Line 136
How was the material deparaffinized? If the automatic deparaffinizing function of the Ventana platform was used, please describe.

Line 144
What magnification objective was used in the scanner (20X or 40X)? Please, describe it.

Line 211
Was any kind of correction of Chi-square test used (e.g. Yates or Pearson)?
I strongly recommend using the Fisher exact test instead of the Chi-square test, as the Fisher test gives an exact result without any correction. Although it can be problematic to use for large numbers of cases, this is not an issue for this study. Also, please provide confidence intervals along with the reported p-values for the presentation of results in the text. Also, after each p value, indicate exactly which test was used to calculate it (you may use an abbreviation, e.g. Kruskal-Wallis: KW, Mann-Whitney U test: MW, etc.).

Table 2
Was there any of the cases in which gastric adenocarcinoma or other malignant tumor was present simultaneously with the GIST?

Line 236
“TNS2 expression in GISTs was relatively close to the level of expression in healthy tissues but overexpressed in GISTs compared with most other cancers (Figure 1; Appendix 1, Figure A1).”
This is not obvious from Figure 1, where the expression of selected non-GIST tumours is similar to normal tissues, but GIST shows higher expression compared to both.
However, the statement is understandable based on Figure A1. This discrepancy should be resolved, e.g. by (also) showing the expression of other normal tissues with higher expression in Figure 1.

Figure 2
As the authors stated in line 153: “Similarly, none of the tumors in the study were graded as weak for DOG1 expression, rendering the DOG1 expression grading -/++/+++. “
However, the figure shows a weak (+) positive DOG1 IHC reaction. Please resolve this discrepancy between the text and Figure 2.

Line 316
It is known that synchronous malignant tumours, especially gastric adenocarcinomas, occur statistically more frequently associated with GISTs than coincidence. The background of this is still not well understood, although attempts have been made to find a common carcinogenetic pathway. In the 148 + 27 GISTs analyzed, was there any evidence of the presence of another tumour in the clinical data, either concurrently with or before/after the diagnosis of the GIST?
Kocsmár et al recently analyzed the synchronous presence of GISTs and gastric adenocarcinomas and found that at least one of the KIT, DOG1 and CD34 IHCs showed ++ or +++ diffuse positivity in GISTs whereas KIT and CD34 were always negative in gastric adenocarcinomas, and DOG1 positivity was found less than half of the cases with weak (+) and usually focal expression. In the light of this, considering the differences in histopathological and immunohistochemical patterns, DOG1 expression of carcinomas does not seem to be a real differential diagnostic problem. Please discuss accordingly and use this as a reference: Kocsmár et al., Cross-testing of major molecular markers indicates distinct pathways of tumorigenesis in gastric adenocarcinomas and synchronous gastrointestinal stromal tumors, Sci Rep 10, 22212 (2020). https://doi.org/10.1038/s41598-020-78232-2.
More problematic may be the expression of DOG1 in certain mesenchymal tumors such as synovial sarcoma (15%), leiomyoma (10%), leiomyosarcoma (4%), angiosarcoma (3%), Ewing sarcoma (3%), malignant peripheral nerve sheath tumor (3%) and schwannoma (1%), which may occur in the same localization as GISTs. (Swalchick et al., Is DOG1 Immunoreactivity Specific to Gastrointestinal Stromal Tumor? Cancer Control. 2015 Oct;22(4):498-504. doi: 10.1177/107327481502200416)

Author Response

We thank the reviewer for their valuable and helpful comments. Point-by-point comments and responses are included below. One additional change was made; Figure 3 has been remade to better represent the distribution of relative gene expressions in our samples.

It is a well-prepared manuscript containing new scientific findings. Their translation into everyday diagnostics is also promising. However, some clarifications are still needed on certain points for publication.

Detailed comments:

Line 105
sarcoma screening series:
It should be described how this differs from the previous series, as it also includes GISTs. Are these also included in the other series or are they separate cases? What criteria were used to select this series?

RE: The following has been added (lines 109-111): “The 27 GISTs included in this series are independent of the GISTs included in the 148-tumor series described above. Patients with a primary tumor who were diagnosed between 1987 and 2012 were included in the study.”

Line 117
What was the reason for not adapting TNS2 IHC as an automated procedure to run on the Ventana immunohistochemistry platform?

RE: We do not have access to the Ventana IHC platform ourselves. The KIT and DOG IHC stainings were conducted with the Ventana IHC platform as a contract service from HUSLAB, Helsinki Central Hospital. TNS2 IHC was performed manually in our research laboratory facilities.

Line 128
“30 minutes at room temperature in the secondary antibody”
in -> with (I suggest to use “with” instead of “in”)

RE: This error has now been fixed.

Line 136
How was the material deparaffinized? If the automatic deparaffinizing function of the Ventana platform was used, please describe.

RE: For the TNS2 IHC, the deparaffinization has been explained on lines 119-122. For the KIT and DOG IHCs, the automatic deparaffinization function of the Ventana platform was used with the EZ-Prep solution. We have added this information to the manuscript as well (lines 139-141)

Line 144
What magnification objective was used in the scanner (20X or 40X)? Please, describe it.

RE: A 20x magnification object was used; this information has been added to the manuscript (line 150)

Line 211
Was any kind of correction of Chi-square test used (e.g. Yates or Pearson)?
I strongly recommend using the Fisher exact test instead of the Chi-square test, as the Fisher test gives an exact result without any correction. Although it can be problematic to use for large numbers of cases, this is not an issue for this study. Also, please provide confidence intervals along with the reported p-values for the presentation of results in the text. Also, after each p value, indicate exactly which test was used to calculate it (you may use an abbreviation, e.g. Kruskal-Wallis: KW, Mann-Whitney U test: MW, etc.).

RE:  All of the Chi-square test results were changed for Fisher-Freeman-Halton Exact tests. Means and 95% CI:s were added to section 3.3. We also added the information of the test used after each p-value as per your suggestion.

Table 2
Was there any of the cases in which gastric adenocarcinoma or other malignant tumor was present simultaneously with the GIST?

RE: Unfortunately, we do not have this information, as medical records concerning other malignancies of the patients were not collected by the biobank.

Line 236
“TNS2 expression in GISTs was relatively close to the level of expression in healthy tissues but overexpressed in GISTs compared with most other cancers (Figure 1; Appendix 1, Figure A1).”
This is not obvious from Figure 1, where the expression of selected non-GIST tumours is similar to normal tissues, but GIST shows higher expression compared to both.
However, the statement is understandable based on Figure A1. This discrepancy should be resolved, e.g. by (also) showing the expression of other normal tissues with higher expression in Figure 1.

RE: More healthy tissues were added to Figure 1 to make the conclusion more convincing. The Appendix A, Figure A1 represents the relative TNS2 expression results in all investigated tissue types, but due to the large number of different tissue types, it is not a reader friendly figure and we decided to present only a subset of the results in the Figure 1.

Figure 2
As the authors stated in line 153: “Similarly, none of the tumors in the study were graded as weak for DOG1 expression, rendering the DOG1 expression grading -/++/+++. “
However, the figure shows a weak (+) positive DOG1 IHC reaction. Please resolve this discrepancy between the text and Figure 2.

RE: As stated on lines 155-159, the individual TMA spots (three replicates per specimen) were graded -/+/++/+++, and in a case of heterogenous staining intensity, the highest degree of expression in the three replicates determined the expression level of that specimen. The figure is an example of weak DOG1 staining in a TMA spot although the sample was graded to be intermediate.

Line 316
It is known that synchronous malignant tumours, especially gastric adenocarcinomas, occur statistically more frequently associated with GISTs than coincidence. The background of this is still not well understood, although attempts have been made to find a common carcinogenetic pathway. In the 148 + 27 GISTs analyzed, was there any evidence of the presence of another tumour in the clinical data, either concurrently with or before/after the diagnosis of the GIST?

RE: Unfortunately, we do not have this information, as medical records concerning other malignancies of the patients were not retrieved by the biobank.

Kocsmár et al recently analyzed the synchronous presence of GISTs and gastric adenocarcinomas and found that at least one of the KIT, DOG1 and CD34 IHCs showed ++ or +++ diffuse positivity in GISTs whereas KIT and CD34 were always negative in gastric adenocarcinomas, and DOG1 positivity was found less than half of the cases with weak (+) and usually focal expression. In the light of this, considering the differences in histopathological and immunohistochemical patterns, DOG1 expression of carcinomas does not seem to be a real differential diagnostic problem. Please discuss accordingly and use this as a reference: Kocsmár et al., Cross-testing of major molecular markers indicates distinct pathways of tumorigenesis in gastric adenocarcinomas and synchronous gastrointestinal stromal tumors, Sci Rep 10, 22212 (2020). https://doi.org/10.1038/s41598-020-78232-2.
More problematic may be the expression of DOG1 in certain mesenchymal tumors such as synovial sarcoma (15%), leiomyoma (10%), leiomyosarcoma (4%), angiosarcoma (3%), Ewing sarcoma (3%), malignant peripheral nerve sheath tumor (3%) and schwannoma (1%), which may occur in the same localization as GISTs. (Swalchick et al., Is DOG1 Immunoreactivity Specific to Gastrointestinal Stromal Tumor? Cancer Control. 2015 Oct;22(4):498-504. doi: 10.1177/107327481502200416)

RE: These points were taken to consideration and have now been discussed more in-depth in the manuscript (lines 343-347).

Reviewer 4 Report

In the current paper the Authors have analyzed the expression of Tensin2 (TNS2) in 176 Gastrointestinal stomal tumor (GIST) and 521 other sarcomas. Results showed a TNS2 intermediate or high expression in 71.4% of the GIST samples. The majority of other sarcomas (89.8%) were negative for TNS2, and intermediate to strong staining was only seen in 2.9% of the samples. Strong TNS2 staining associated with a gastric location (gastric 52.8% vs. non-gastric 7.2%; P<0.001), an absence of metastases (non-metastatic tumors 44.3% vs. metastatic tumors 5.9%; P=0.008), female sex (female 45.9% vs. male 33.8%; P=0.027) and tumors of lower risk categories (very low or low 46.9% vs. intermediate 51.7% vs. high 29.0%; P=0.025). Interestingly, TNS2 expression did not correlate with overall survival or metastasis-free survival. No associations between TNS2 expression and tumor size, mitotic count, or age of the patient were detected. In conclusion, these data suggest evidence for the value of TNS2 as a sensitive and specific diagnostic biomarker for GIST.

In my opinion, the overall level of the paper is very good: it is well written and some important considerations are highlighted. The procedures appear suitable and the conclusions seem appropriate and rationale, emphasizing the possibility to introduce this marker in the diagnostic algorithm of GIST.

Some minor changes should be performed:

1.     If is possible improve the quality of microphotos.

2.     Please check the few spelling grammatical error in the text.

Author Response

We thank the reviewer for their valuable and helpful comments. Point-by-point comments and responses are included below. One additional change was made; Figure 3 has been remade to better represent the distribution of relative gene expressions in our samples.

In the current paper the Authors have analyzed the expression of Tensin2 (TNS2) in 176 Gastrointestinal stomal tumor (GIST) and 521 other sarcomas. Results showed a TNS2 intermediate or high expression in 71.4% of the GIST samples. The majority of other sarcomas (89.8%) were negative for TNS2, and intermediate to strong staining was only seen in 2.9% of the samples. Strong TNS2 staining associated with a gastric location (gastric 52.8% vs. non-gastric 7.2%; P<0.001), an absence of metastases (non-metastatic tumors 44.3% vs. metastatic tumors 5.9%; P=0.008), female sex (female 45.9% vs. male 33.8%; P=0.027) and tumors of lower risk categories (very low or low 46.9% vs. intermediate 51.7% vs. high 29.0%; P=0.025). Interestingly, TNS2 expression did not correlate with overall survival or metastasis-free survival. No associations between TNS2 expression and tumor size, mitotic count, or age of the patient were detected. In conclusion, these data suggest evidence for the value of TNS2 as a sensitive and specific diagnostic biomarker for GIST.

In my opinion, the overall level of the paper is very good: it is well written and some important considerations are highlighted. The procedures appear suitable and the conclusions seem appropriate and rationale, emphasizing the possibility to introduce this marker in the diagnostic algorithm of GIST.

Some minor changes should be performed:

  1. If is possible improve the quality of microphotos.

RE: For some reason, the quality of the pictures in the manuscript was much lower than our high resolution figures that we submitted. We will be in contact with the editor about this issue.

  1. Please check the few spelling grammatical error in the text.

RE: Any grammatical errors found were fixed.

Reviewer 5 Report

Major comment:
The authors state in the abstract that “TNS2 was found to be downregulated in most human cancers and overexpressed in GIST...”
While they analyze IST transcriptome data, including publicly available tumor data, such as the TCGA, cBioPortal, etc., would make their argument much more compelling, and, probably, statistically significant as well.

The authors should analyze tumor protein and transcript levels, present these public data for pan-cancer, gastric cancers, and gastrointestinal tumors.

With appropriate statistical modeling, and cut-offs, they should present data to a slew of questions like
a) TNS2 expression in cancers?
b) TNS2 expression in GCs?

c) TNS2 expression in GISTs?
Comparing a, b, and c, how far/near is the TNS2 level to the median protein/transcript expression relative to normal/benign tissue?

This will also give confidence to the use of TNS2 as biomarker in clinic.

Minor Comments:

The importance of TNS2 in GISTs is presented in this study; however other studies (and reviews) have shed light on the role of this protein in other cancer types, which should be at least briefly discussed. Not only will this shed light on the role of TNS2 in general, but also highlight its relative importance in GISTs. For eg. The role of Tensins, and their importance in gastric cancers has been presented by Nizol et. al. (PMID 33926026).

Line 15 (Abstract): should it be “... most human cancers but overexpressed in GIST” instead of “and”? 

Author Response

We thank the reviewer for their valuable and helpful comments. Point-by-point comments and responses are included below. One additional change was made; Figure 3 has been remade to better represent the distribution of relative gene expressions in our samples.

Major comment:
The authors state in the abstract that “TNS2 was found to be downregulated in most human cancers and overexpressed in GIST...”
While they analyze IST transcriptome data, including publicly available tumor data, such as the TCGA, cBioPortal, etc., would make their argument much more compelling, and, probably, statistically significant as well.

The authors should analyze tumor protein and transcript levels, present these public data for pan-cancer, gastric cancers, and gastrointestinal tumors.

With appropriate statistical modeling, and cut-offs, they should present data to a slew of questions like
a) TNS2 expression in cancers?
b) TNS2 expression in GCs?

  1. c) TNS2 expression in GISTs?
    Comparing a, b, and c, how far/near is the TNS2 level to the median protein/transcript expression relative to normal/benign tissue?

This will also give confidence to the use of TNS2 as biomarker in clinic.

RE: We agree that the study would benefit from the use of public databases. We made an effort to investigate TNS2 expression with publicly available databases and tools, but GIST gene expression profiles were not available in harmonized or curated datasets. Therefore, the suggested analyses would require an enormous bioinformatic work to collect and harmonize data from various sources and is beyond our resources.

Minor Comments:

The importance of TNS2 in GISTs is presented in this study; however other studies (and reviews) have shed light on the role of this protein in other cancer types, which should be at least briefly discussed. Not only will this shed light on the role of TNS2 in general, but also highlight its relative importance in GISTs. For eg. The role of Tensins, and their importance in gastric cancers has been presented by Nizol et. al. (PMID 33926026).

RE: We have now added more discussion on the importance of TNS2 in gastric cancer (lines 399-404) as per your suggestion.

Line 15 (Abstract): should it be “... most human cancers but overexpressed in GIST” instead of “and”? 

RE: The error has now been fixed.

Round 2

Reviewer 2 Report

I think the authors have made a good revision and their answers for important comments are well presented.

Reviewer 5 Report

The authors' reply is valid. Mining public databases will, no doubt, require bioinformatician expertise and continuous and constant collaboration, which may indeed be very expensive.
Nevertheless, the confidence in the data as presented at present is very low - more so, if the claim being made is that a given protein has differential/diagnostic biomarker potential.